# Modulating Tumor Immunity by Targeting Tumor Fibrotic Stroma and Angiogenic Vessels for Lung Cancer Treatment

**DOI:** 10.3390/cancers16132483

**Published:** 2024-07-08

**Authors:** Yi Yuan, Falguni Mishra, Bin Li, Guangda Peng, Payton Chan, Jenny Yang, Zhiren Liu

**Affiliations:** 1Department of Biology, Georgia State University, Atlanta, GA 30303, USA; yyuan13@gsu.edu (Y.Y.); fmishra1@student.gsu.edu (F.M.); bli17@student.gsu.edu (B.L.); gpeng1@gsu.edu (G.P.); pchan2@student.gsu.edu (P.C.); 2Department of Chemistry, Georgia State University, Atlanta, GA 30303, USA; jenny@gsu.edu

**Keywords:** cancer-associated fibroblasts, tumor vessels, tumor immunity, integrin α_v_β_3_, checkpoint blockade, immunotherapy

## Abstract

**Simple Summary:**

Yuan et al. present a strategy to modulate tumor immunity by simultaneously depleting CAFs and tumor angiogenic vessels using a rationally designed protein that induces integrin α_v_β_3_-expressing cell apoptosis. The study offers a unique opportunity for the enhancement of cancer immunotherapies, especially for patients with a tumor of dense stroma and high angiogenesis.

**Abstract:**

Fibrotic stroma and angiogenic tumor vessels play an important role in modulating tumor immunity. We previously reported a rationally designed protein (ProAgio) that targets integrin α_v_β_3_ at a novel site. ProAgio induces the apoptosis of cells that express high levels of the integrin. Both activated cancer-associated fibroblasts (CAFs) and angiogenic endothelial cells (aECs) in tumors express high levels of integrin α_v_β_3_. ProAgio simultaneously and specifically induces apoptosis in CAFs and aECs in tumors. We provide evidence here that the depletion of CAFs and the elimination of leaky tumor angiogenic vessels by ProAgio alter tumor immunity. ProAgio reduces CD_4_^+^ Treg and Myeloid-derived suppressor cells (MDSCs), increases CD_8_^+^ T-cells, and increases the M1/M2 macrophage ratio in the tumor. The depletion of dense fibrotic stroma (CAFs) by ProAgio decreases the *Programmed Death Ligand 1* (*PDL-1*) levels in the stroma areas surrounding the tumors, and thus strongly increases the delivery of anti-PDL-1 antibody to the target cancer cells. The impact of ProAgio on tumor immunity provides strong synergistical effects of checkpoint inhibitors on lung cancer treatment.

## 1. Introduction

Lung cancer is often believed to be more immunogenic with high levels of PDL-1 expression in tumors [1,2]. Thus, checkpoint inhibitors are regarded as a promising strategy both as a single agent and in combination with other chemotherapeutics. Due to relatively good clinical outcomes, the checkpoint inhibitor and the combination therapies have been established as standard-of-care treatments for lung cancers [3,4,5]. However, despite the success of checkpoint blockade immunotherapies in lung cancer treatment [6], patients did not fully benefit from the treatments, due to different resistance mechanisms [7,8,9]. It has been observed that many patients do not respond initially to treatment [10]. It was also noted that even if patients initially responded to the checkpoint blockade treatments, the disease often relapses after a period of response due to the development of immunotherapy resistance [11,12]. It has been gradually realized that the tumor immune escape and, also, the resistance to immune checkpoint blockade therapies do not depend solely on the immunogenic potential of cancer cells but rather mainly receive contributions from the tumor stroma. The tumor microenvironment (TME) plays an important role in educating the host immuno-responses in favor of tumor progression. Among all components, tumor angiogenesis and cancer-associated fibroblasts (CAFs) are particularly important in the tumor’s immunosuppression adoptive program [13,14,15].

CAFs are the most prominent cell types in TME, which actively engage in crosstalk with the surrounding cells to promote cancer cell proliferation and survival. CAFs are a central player in cancer treatment resistance including resistance to immunotherapies [16,17]. The role played by CAFs in tumor immunity is fourfold: (1) they orchestrate the dense fibrotic stroma that forms a physical barrier for immune cell infiltration into the tumor, (2) they are the predominate source of immunosuppressive molecules, such as cytokines/chemokines and growth factors, which suppress immune cell activation, (3) CAFs help cancer cells to express high levels of immune checkpoint blockades such as PDL-1, and (4) activated CAFs form a physical trap for checkpoint blockades such as PDL-1 antibody due to the high-level expression of PDL-1 on CAFs.

Tumor angiogenic vessels are another critical player in modulating tumor immunity. High tumor microvasculature closely correlates with a poor prognosis in cancer patients. Tumor angiogenesis modulates tumor immunity in the following ways. (1) The overexpression of several angiogenesis-stimulating growth factors, such as vascular growth factors (VEGFs) and platelet derived growth factors (PDGFs), can directly activate immunosuppressive cells and suppress immune effector cells to shape the immunosuppressive microenvironment [18,19]. (2) While the abnormal vascular structure of tumors prevents immune cell infiltration through a selective immune cell barrier or inhibit immune cell function by exacerbating hypoxia microenvironment [20], meanwhile, activated immunosuppressive cells can further promote abnormal angiogenesis [21]. (3) Leaky and disorganized tumor vessels often lead to high hypoxia in tumor. Hypoxia-inducible factor Hif-1a is known to be an important immunosuppressive factor [22,23].

Both aECs and CAFs in tumors express high levels of integrin α_v_β_3_ [24,25]. We previously reported a rationally designed protein, ProAgio, that targets integrin α_v_β_3_ at a novel site and induces apoptosis in the integrin-expressing cells by recruiting caspase 8 at the cytoplasmic domain of β_3_ [26]. We report here that ProAgio induces apoptosis in integrin α_v_β_3_-expressing CAFs and aECs in lung tumors. The depletion of CAFs by ProAgio decreases intratumoral collagen. The depletion of CAFs by ProAgio consequentially decreases tumor immunosuppressive effects, e.g., an increase in CD8^+^ T-cells and a decrease in CD4^+^ Treg cells and MDSCs in lung tumors. ProAgio also increases the macrophage M1/M2 ratio in tumors. In addition, the anti-angiogenetic effect of ProAgio eliminates angiogenic leaky tumor vessels, which consequently facilitates immune cell infiltration and decreases hypoxia in murine models of lung cancers. The depletion of CAFs by ProAgio reduces immune checkpoint molecule PDL-1 levels in tumor stroma areas due to the high-level expression of PDL-1 on CAFs. In addition, the depletion of CAFs by ProAgio prevents checkpoint blockade from being trapped by CAFs due to the high expression of PDL-1 on CAFs. Because of the effects of ProAgio in modulating tumor immunity by simultaneously depleting CAFs and eliminating tumor angiogenic vessels, a synergistic treatment effect of ProAgio in combination with checkpoint blockade aPDL-1 is observed, suggesting an excellent treatment strategy for lung cancer patients, especially for the patients whose tumors exhibit resistance to and have relapsed from checkpoint blockade immunotherapies.

## 2. Materials and Methods

### 2.1. Cell Line and Human CAFs

The lung adenocarcinoma cell line A549 was purchased from ATCC. Cells were cultured in Dulbecco’s Modified Eagle Medium (DMEM) under a humidified atmosphere of 5% CO_2_ at 37 °C, supplemented with 10% fetal bovine serum and 1% penicillin. CAFs isolated from patients were purchased from Neuromics. The cells were cultured according to the vendor’s suggestion.

### 2.2. Genetically Engineered Lung Cancer Mouse Model (GEM-NSCLC) and Treatments

All animal experiments were conducted following NIH guidance and approved by the Institutional Animal Care and Use Committee of Georgia State University. A549 xenograft: A540 (5 × 10^5^) cells were implanted into the right flank of nude mice. Tumor growth was measured using a caliper ruler. For the GEM-NSCLC model, 10 to 12-week-old KP mice (LSL-KrasG12D/+; LSL-Trp53R172H/+) were utilized. The breeding offspring mice were genotyped for the correct KrasLSL-G12D and p53 R172H/fl alleles by PCR using the following PCR primer pairs and following protocols from the Jackson Laboratory website (http://web.mit.edu/jacks-lab/protocols_table.html, accessed on 1 July 2024): genotyping PCR primer pairs P53F(5′-AGCCTGCCTAGCTTCCTCAGG-3′) and P53R(5′CTTGGAGACATAGCCACACTG-3′), or KRAS R(5′-TGTCTTTCCCCAGCACAGT-3′), KRAS F(5′-GCAGGTCGAGGGACCTAATA-3′) and KRAS WT(5′-CTGCATAGTACGCTATACCCTGT-3′). To induce recombination, a replication-deficient adenovirus-expressing Cre (AdV-Cre) from the University of Iowa Gene Transfer Vector Core was delivered via intratracheal intubation. The titer of AdV-Cre was 5.0 × 10^7^ PFU/mL and the total volume of AdV-Cre was 75 μL per mouse. The described treatments began 12 weeks after AdV-Cre delivery. Mice were randomly assigned to treatment groups, with early treatment (ET) consisting of 15 mg/kg ProAgio for 4 consecutive days and late treatment (LT) involving the same dosage for 12 days. Survival treatment included 15 mg/kg ProAgio administered for 10 consecutive days, followed by 10 doses every other day, 4 mg/kg InVivoMAb anti-mouse PD-L1 (B7-H1) (#BE0101, Bioxcell (Lebanon, NH, USA)) twice per week for 4 weeks or a combination. At the end of treatments, mice were either sacrificed for analysis or maintained in cages for survival assessment. Organs, tumor tissues, and blood samples were collected for subsequent analyses. Statistical comparisons were made against the control group.

### 2.3. Tissue Staining: IHC, Sirius Red, and Immunofluorescence (IF)

IHC, Sirius red, and IF staining procedures were similar to those of previous reports [27,28]. The quantification of Sirius red, IHC, and immunofluorescence staining was conducted utilizing ImageJ 1.54d software. The quantities are presented as described in figure legends.

### 2.4. Cell Apoptosis Assay, Immune Cell FACS Analyses, and Hydroxyproline Assay

The procedures for these assays were similar to our previous reports [28,29]. FACS data were analyzed by FlowJo v10.9 software (Tree Star (Ashland, OR, USA)).

### 2.5. Cytokine and Chemokine ELISA Quantification Assay

For quantifying chemokines and cytokines, the lung tissues from tumor-bearing mice before and after treatment were collected and analyzed using ELISA kits according to the manufacturer’s instructions. Calibrator blends provided in the kits were added and analyzed together with the samples on the same plates. The concentrations of calibrator blends and raw values obtained after reading the plates were used to generate standard curves to calculate the concentrations of each measurement [26].

### 2.6. Statistical Analyses

Statistical analyses were performed using GraphPad Prism 9.0 software, and each experiment was conducted a minimum of three times. Significance was assessed using Student’s *t*-test and/or one-way ANOVA for multiple comparisons. In all figures, the statistical significance level was defined as *p* < 0.05; n.s. indicates not significant. All data are presented as mean ± SEM.

## 3. Results

### 3.1. ProAgio Enhances Checkpoint Inhibitor Immunotherapy in Lung Cancer

We previously reported the depletion of CAFs and the elimination of tumor angiogenesis in tumors of breast and pancreatic cancers by ProAgio, a rationally designed protein that targets integrin α_v_β_3_ [26,27,28]. Lung cancers, particularly NSCLC, are often highly angiogenic [30] and are frequently associated with the activation of CAFs and abundant stroma [31]. We first probed integrin a_v_b_3_ levels in lung cancer CAFs and found that the integrin was highly expressed in commercially available lung CAFs isolated from the tumor of lung cancer patients (Figure 1A). We therefore reasoned that ProAgio would be an excellent drug candidate for lung cancer treatment. To test our speculation, we first examined the effects of ProAgio on the CAFs isolated from lung cancer patient tumors. Clearly, ProAgio induced the apoptosis of the CAFs (Figure 1B). We next employed a s.c. xenograft model of human NSCLC A549 cells. ProAgio inhibited A549 tumor growth (Figure 1C). ProAgio treatment decreased tumor angiogenesis (Figure 1D). Furthermore, the IHC of a-SMA, Sirius red staining, and hydroxyproline assay showed that ProAgio treatment reduced collagen and α-SMA-positive cells in the tumors (Figure 1E–G). To further test whether ProAgio is effective in lung cancer treatment, we employed a genetically engineered mouse (GEM) NSCLC model (Ref to as GEM-NSCLC) generated from breeding Kras^G12D^ and P53^R172H^ mice, followed by an intratracheal intubation delivery of adenoviral coded Cre recombinase (AdV-Cre) [32]. The GEM mice were treated with vehicle or ProAgio (15 mg/kg, 15 Q1D doses) 126 days after AdV-Cre delivery (Figure 2A). The mice were maintained in their home cages for assessment of survival. Evidently, ProAgio treatment greatly increased the survival of tumor-bearing mice (Figure 2B). Randomly selected mice (*n* = 9) were euthanized 10 days after the last dose of treatment. The close examination of the lungs of the treated mice revealed that ProAgio reduced the lung tumor nodules both in size and number in the GEM mice, especially the large-size nodules (Figure 2C–E). The Sirius Red and IHC of a-SMA staining demonstrated that ProAgio depleted activated CAFs and reduced tumor collagen levels (Figure 2F–H). The IHC staining of CD31 with the tumor section showed thatProAgio reduced tumor angiogenic vessels (Figure 2I,J). Our results suggest that ProAgio simultaneously targets CAFs and tumor angiogenesis and is potentially a drug candidate for lung cancer treatment.

Due to clinical success, checkpoint inhibitors immunotherapies alone and in combination with chemotherapies have been established as standard-of-care treatments for lung cancers [8]. We therefore questioned whether ProAgio would enhance checkpoint inhibitor immunotherapy in lung cancer. To this end, we employed the GEM-NSCLC model. The GEM mice were treated with vehicle, ProAgio (15 mg/kg, 10 Q1D plus 10 Q.O.D. doses), anti-PDL-1 antibody (aPDL-1, 4 mg/kg, twice weekly for four weeks), and ProAgio + aPDL1 12 weeks after the delivery of AdV-Cre (Figure 3A). ProAgio or αPDL-1 alone provided survival advantage, while ProAgio + αPDL-1 provided an even greater increase in survival benefits (tumors in more than half of the mice in the ProAgio + αPDL-1 group disappeared) (Figure 3B). ProAgio reduced the lung tumor nodules both in size and number in the GEM mice. ProAgio + aPDL1 almost completely eliminated the tumor nodules in the lungs of the treated mice, particularly the large-size nodules (Figure 3C–E). ProAgio treatment depleted CAFs and reduced collagen in tumors. Interestingly, ProAgio + aPDL1 further decreased CAFs and collagen in tumors compared to the ProAgio alone group (Figure 3F–H). We speculate that aPDL-1 might lead to the apoptosis of CAFs in the tumor due to the block of PDL-1 on the activated CAFs. The staining of CD31 demonstrated that ProAgio eliminated angiogenic tumor vessels. The ProAgio and aPDL-1 combination led to more reduction in tumor angiogenic vessels (Figure 3I,J).

### 3.2. ProAgio Alters Lung Tumor Immunity

The depletion of CAFs and the abrogation of tumor angiogenesis alter tumor immunity. To test whether ProAgio treatment that depletes CAFs and eliminates angiogenic vessels affects tumor immunity, GEM-NSCLC mice were treated with either 4 daily doses (referred to as early treatment or ET) or 12 daily doses (referred to as late treatment or LT) of ProAgio or vehicle 12 weeks after the delivery of AdV-Cre. Animals were euthanized two days following the last dose (Figure 4A). Similarly, the Sirius red and IHC of a-SMA staining indicated the reduction in collagen and CAFs in the tumors and surrounding lung tissues in both ET and LT treatments. CD31 staining demonstrated a decrease in tumor angiogenesis. Hypoxia strongly affects immunity [22,23]. The IHC staining of tumor sections suggested that ProAgio treatment decreased Hif-1a in lung tumors (Figure 4B,C), suggesting that, similar to our observations with pancreatic and breast cancer, ProAgio treatment decreased lung cancer hypoxia due to its anti-angiogenic effects. Immune cells in tumors were subsequently analyzed via FACS. ProAgio treatment resulted in a >2-fold and >3-fold increase in CD8^+^ T-cells in the ET and LT groups, respectively (Figure 4D). ProAgio treatment led to a >3-fold and >10-fold decrease in CD4^+^ Treg cells in the ET and LT groups, respectively (Figure 4E). Furthermore, ProAgio treatment also decreased MDSCs (Figure 4F) and increased the M1 to M2 macrophage ratio in tumors (Figure 4G). Overall, ProAgio strongly increased anti-tumor immunity and reduced cancer immune resistance. The effects were also reflected by the changes in cytokine/chemokine profiles in the lungs of the tumor-bearing mice. IL-6, CXCL2, and CXCL12 are cytokines/chemokines that are released by CAFs to regulate tumor immunity and angiogenesis. ProAgio treatment altered IL-6, CXCL2, and CXCL12 levels in the lungs (Figure 4H–J).

### 3.3. Elimination of CAFs and Leaky Angiogenic Vessels by ProAgio Increased the Delivery of aPDL-1 to Lung Tumors, Particularly in the Histological Carcinoma Regions

We previously reported that the depletion of CAFs and the elimination of tumor leaky vessels enhanced drug (small- and large-molecule) delivery to tumors [27,28]. It is well known that CAFs express high levels of PDL-1 [33]. Thus, a dense fibrotic stroma forms both a physical barrier and a molecular trap for the administered aPDL-1, preventing it from reaching its target cancer cells. The depletion of the fibrotic stroma and the elimination of leaky angiogenic vessels by ProAgio would facilitate the delivery of aPDL-1 to cancer cells in lung tumors. To confirm whether the depletion of the fibrotic stroma and angiogenic vessels indeed helps the administered aPDL-1 to reach the target cancer cells, we first analyzed PDL-1 in the lungs and lung tumor nodules of treated GEM-NSCLC mice. Clearly, PDL-1 levels in the lung of tumor-bearing mice decreased by approximately 2-fold, while the PDL-1 levels in the tumor nodules increased by approximately 1.4-fold, upon ProAgio treatment (Figure 5A–C). The observation suggests that ProAgio depletes high-PDL-1-expression-activated CAFs in the tumor-containing lung, thereby leading to a decrease in PDL-1 levels. We do not fully understand why ProAgio leads to an increase in PDL-1 within the tumor nodules. However, an increase in PDL-1 levels upon ProAgio treatment would be consistent with the strong synergistic effects of ProAgio in combination with aPDL-1. We next analyzed the levels of the administered aPDL-1 and the localization of the aPDL-1 relative to a-SMA positive CAFs and epithelium carcinoma cancer cells 12 h after administration. Evidently, the administered aPDL-1 levels were high in the lung tissue surrounding the tumor nodules, while the aPDL-1levels inside the tumor nodules were very low in the vehicle treated group. Conversely, the administered aPDL-1 levels were relatively low in the lung tissue surrounding the tumor nodules, while the aPDL-1levels inside the tumor nodules were very high in the ProAgio-treated group (Figure 5D–F). Thus, we conclude that the depletion of CAFs by ProAgio in the tumor fibrotic stroma reduced PDL-1 levels in the stroma, which consequently prevented the administered aPDL-1 from being trapped in the tumor microenvironment mainly by CAFs, facilitating the accumulation of more administered antibody in the carcinoma areas. This action will certainly enhance the effectiveness of aPDL-1.

## 4. Discussion

Lung cancers, particularly advanced disease, usually have high levels of activated CAFs and are rich in angiogenic vessels. The highly activated CAFs and angiogenesis orchestrate lung cancer progression and resistance to drug treatment. The cancer-promoting CAFs and angiogenesis modulate tumor immunity. It is well established that activated CAFs and tumor angiogenesis contribute to checkpoint blockade immunotherapy resistance in lung cancer [13,14,15]. Furthermore, lung cancer is often associated with high levels of PDL-1 expression in tumors [1,2]. PDL-1 is not only expressed on lung cancer cells but also on the CAFs in the tumor. The high level of PDL-1 on the CAFs forms a molecular trap to prevent PDL-1 blockade from reaching the target cancer cells, thus reducing the treatment effectiveness of the blockade. Therapeutic strategies targeting the tumor fibrotic stroma have been actively explored, particularly in the treatment of pancreatic ductal adenocarcinoma (PDAC) due to the unique properties of desmoplasia of PDAC. However, there is a very limited success [34,35]. Due to the important role of CAFs and angiogenesis in modulating tumor immunity, targeting CAFs and tumor angiogenesis to improve checkpoint blockade immunotherapy in lung cancer treatment has become a promising treatment strategy. We provide an example of simultaneously depleting lung cancer CAFs and eliminating angiogenic vessels by a novel rationally designed protein drug targeting integrin a_v_b_3_ [27,28]. The depletion of CAFs and the elimination of leaky tumor angiogenic vessels by ProAgio alter tumor immunity as revealed by an increase in CD8^+^ T-cells and decreases in CD4^+^ Treg and MDSCs in the tumor. ProAgio also changes tumor immunity through an increase in the M1/M2 macrophage ratio. Due to the effect of ProAgio on tumor immunity, the drug strongly enhances treatment effectiveness with PDL-1 blockade aPDL-1. We show here that ProAgio enhances aPDL-1 treatment effectiveness in the following ways (Figure 5G): (1) the alteration of tumor immunity due to the depletion of CAFs and tumor angiogenesis, (2) the increase in tumor blood perfusion, and thus an increase in the infiltration of aPDL-1 due to the elimination of leaky and disorganized angiogenic tumor vessels, (3) the enhancement of anti-tumor immunity due to a decrease in tumor hypoxia, and (4) the removal of the molecular trap of aPDL-1 due to the depletion of activated CAFs. Although checkpoint blockade immunotherapy with or without combination with chemotherapies has become a standard-of-care regimen for lung cancer patients, a large percentage of patients do not respond initially to treatment or relapse after a period of response mainly due to the immunosuppressive effects of activated CAFs and angiogenesis in the tumors. Thus, the simultaneous depletion of angiogenic vessels and activated CAFs by ProAgio will certainly provide very important benefits in lung cancer treatment, especially in combination with checkpoint blockade immunotherapy.

ProAgio decreased PDL-1 in the lung, most likely due to the depletion of PDL-1-expressing CAFs and aECs. However, the treatment increased PDL-1 in the tumor nodules, which is consistent with the observation that ProAgio in combination with aPDL-1 led to such a strong enhancement of effectiveness in the treatment. It is well known that chemotherapies upregulate PDL-1 expression on cancer cells due to the induction of apoptosis of cancer cells [36]. Thus, it is plausible that the depletion of activated CAFs and angiogenic tumor vessels by ProAgio consequently results in cancer cell apoptosis, and thus increases PDL-1 levels on cancer cells. The increase in PDL-1 levels in lung tumor nodules would certainly increase the accumulation of administered aPDL-1. However, the increase in aPDL-1 levels is not proportional to the increase in PDL-1 levels in the tumor nodules. Thus, it is less likely that the increase in aPDL-1 upon ProAgio treatment in the tumor nodules is solely due to the increase in PDL-1 levels. Preventing aPDL-1 from being trapped by activated CAFs and angiogenic vessels plays an important role in the increase in aPDL-1 in tumor nodules. How ProAgio and aPDL-1 further decreased CAFs and angiogenesis in the tumor is an open question. A plausible explanation is that aPDL-1 blocked PDL-1 on CAFs and aECs, while ProAgio enhanced inflammation immunity. Thus, the increased infiltration of immune killer cells leads to an increased induction of apoptosis of CAFs and aECs.

## 5. Conclusions

ProAgio, an integrin α_v_β_3_ targeting agent, modulates tumor immunity by depletion of activated cancer associated fibroblasts and tumor angiogenic vessels. The activity of ProAgio in removal of immunosuppressive effects imposed by tumor stroma led to a strong synergistic enhancement to the efficacy of immune checkpoint blockades in lung cancer treatment. Thus, ProAgio offers a unique opportunity to lung cancer treatment, especially for patients with disease that resistance to chemo and immune therapies.

## Figures and Tables

**Figure 1 cancers-16-02483-f001:**
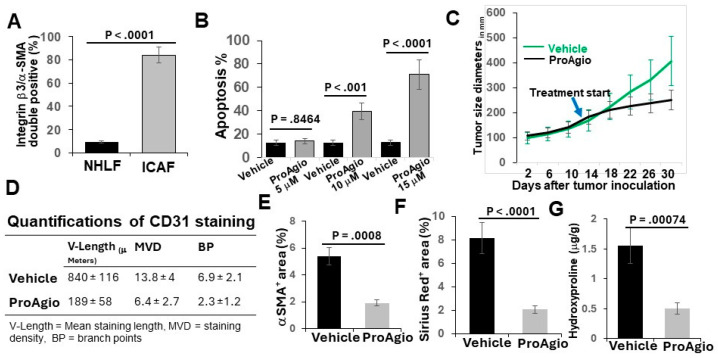
ProAgio depletes CAFs and angiogenic tumor vessels in A549 cell xenograft model. (**A**) Quantification of co-IF staining of integrin β3 and α-SMA in cultured normal human lung fibroblasts (NHLFs) and commercially available lung CAFs (lCAFs) isolated from lung cancer patient tumors. The quantity is presented as integrin β3/α-SMA double positive cell population (%). (**B**) Apoptosis of the CAFs under treatment of different concentrations (indicated) of ProAgio was measured using an apoptosis kit. Cell apoptosis is presented as the percentage of apoptosis by defining the apoptosis of untreated cells as 0%. (**C**) Mean tumor volume (n = 10) of s.c. xenograft of A549 cells. The tumor-bearing mice were treated with indicated agents. The arrow indicates the treatment starting time point. (**D**) Quantifications (n = 6) of CD31 staining of A549 tumor sections. (**E**–**G**) Quantitative analyses of IHC staining of a-SMA (**E**), Sirius red (**F**) staining, and hydroxyproline assay (**G**) in tumor sections/tissues of A549 xenograft mice treated with vehicle or ProAgio. Error bars in (**A**–**G**) represent means ± SEM.

**Figure 2 cancers-16-02483-f002:**
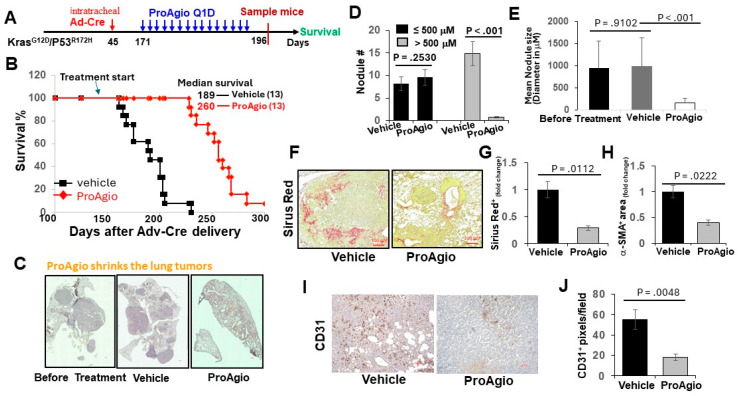
ProAgio shrank lung tumors by depleting activated CAFs and angiogenic tumor vessels. (**A**) The scheme illustrates the ProAgio or vehicle treatment regimen of GEM-NSCLC mice. (**B**) Kaplan–Meier survival analysis of GEM-NSCLC mice treated with vehicle or 15 mg/kg ProAgio. Treatment was started at the indicated time (~179 days of age or 126 days of AdV-Cre delivery). The numbers in parentheses indicate the group size. (**C**) Representative images of lungs of treated mice (treated with indicated agents, scale bar 100 μm). (**D**,**E**) Quantitative measurements (both size (**D**) and # (**E**)) of tumor nodules in the lungs of the treated mice (treated with indicated agents). (**F**,**G**) Representative images (**F**) and quantifications (**G**) of Sirius red staining of lung sections from the treated mice (scale bar in F 100 μm). (**H**) Quantifications of IHC of a-SMA staining of lung sections from the treated mice. The quantity in (**F**–**H**) is presented as fold changes with the Sirius red and α-SMA levels of vehicle treated groups as reference 1. (**I**,**J**) Representative images (**I**) and quantifications (**J**) of IHC of CD31 staining of lung sections from the treated mice (scale bar in I 100 μm). The quantity is presented as CD31^+^ pixels per view field. Error bars in (**D**–**J**) represent means ± SEM.

**Figure 3 cancers-16-02483-f003:**
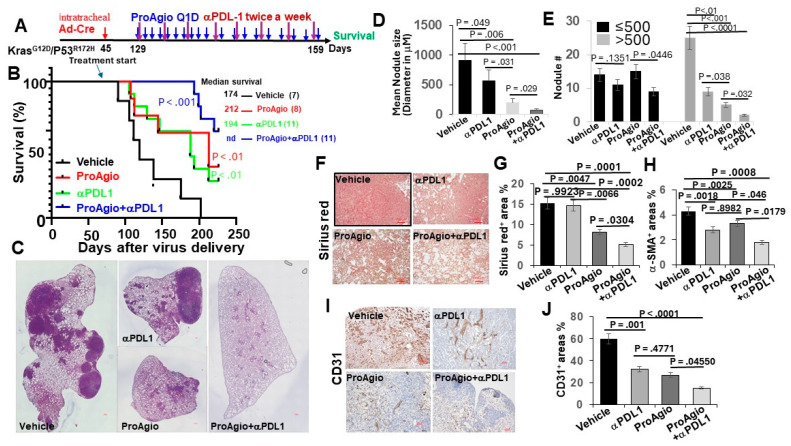
ProAgio synergistically enhances the effects of immune checkpoint blockade therapies. (**A**) The scheme illustrates the ProAgio, aPDL1, and ProAgio + aPDL1 treatment regimen of GEM-NSCLC mice. (**B**) Kaplan–Meier survival analysis of GEM-NSCLC mice treated with indicated agents. Treatment was started at the indicated time (~129 days of age or 12 weeks of AdV-Cre delivery). The numbers in parentheses indicate the group size. (**C**) Representative images of lungs of treated mice (treated with indicated agents, scale bar 100 μm). (**D**,**E**) Quantitative measurements (both size (**D**) and # (**E**)) of tumor nodules in the lungs of the treated mice (treated with indicated agents). (**F**,**G**) Representative images (**F**) and quantifications (**G**) of Sirius red staining of lung sections from the treated mice (scale bar in F 100 μm). (**H**) Quantifications of IHC of a-SMA staining of lung sections from the treated mice. The quantity in (**G**,**H**) is presented as Sirius red^+^ or a-SMA^+^ area (%). (**I**,**J**) Representative images (**I**) and quantifications (**J**) of IHC of CD31 staining of lung sections from the treated mice (scale bar in I 100 μm). The quantity is presented as CD31^+^ area (%). Error bars in (**D**–**J**) represent means ± SEM.

**Figure 4 cancers-16-02483-f004:**
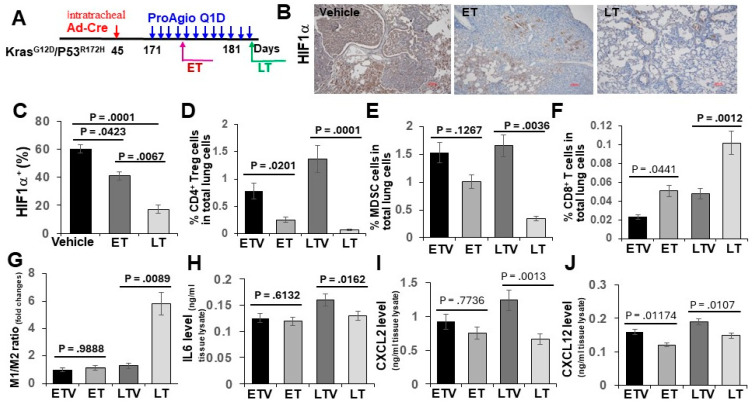
ProAgio alters lung tumor immunity. (**A**) The scheme illustrates the ProAgio and vehicle treatment regimen of GEM-NSCLC mice. (**B**,**C**) Representative images (**B**) and quantifications (**C**) of IHC Hif1a staining of lung sections from the treated mice (scale bar in B 100 μm). Quantification (**C**) is presented as Hif1a^+^ area (%). (**D**–**G**) FACS analyses of CD4^+^ Treg cells (**D**), MDSCs (**E**), CD8^+^ T cells (**F**), and the M1/M2 macrophage ratio (**G**) in lung tissues from mice in the indicated treatment. ETV—early treatment vehicle, ET—early ProAgio treatment, LTV—late treatment vehicle, LT—late ProAgio treatment. Quantities are presented as % of CD4^+^ Treg cells, CD8^+^ T cells, and MDSCs in total lung cells. The M1/M2 macrophage ratio is presented as fold changes against the reference of the ETV M1/M2 ratio set as 1. (**H**–**J**) ELISA analyses of IL6 (**H**), CXCL2 (**I**), and CXCL12 (**J**) in lung tissues from mice in the indicated treatment. The quantities of cytokines/chemokines are presented as ng per ml of lung extracts. Error bars in (**C**–**J**) represent means ± SEM.

**Figure 5 cancers-16-02483-f005:**
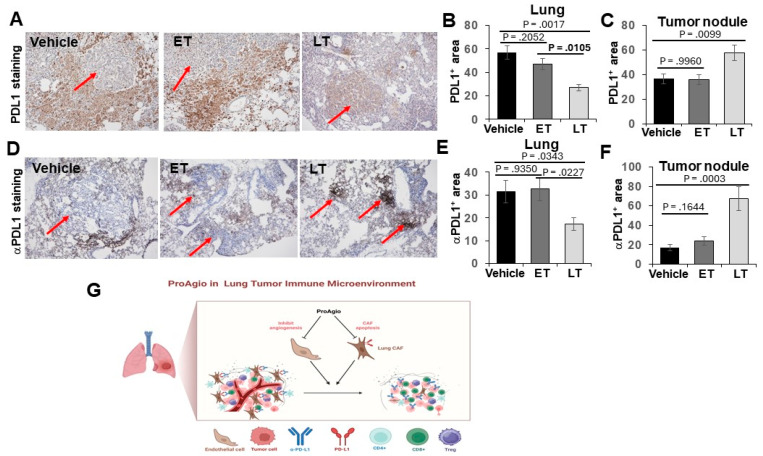
ProAgio increases αPDL1 delivery to lung cancer nodules. (**A**–**C**) Representative images (**A**) and quantifications of IHC PDL1 staining in lung tissue (**B**) and in tumor nodules (**C**) from the treated mice. Quantifications in (**B**,**C**) are presented as PDL1^+^ area (%). (**D**,**F**) Representative images (**D**) and quantifications of IHC αPDL1 staining in lung tissue (**E**) and in tumor nodules (**F**) from the treated mice (scale bar in A & D 100 μm). Quantifications in (**E**,**F**) are presented as αPDL1^+^ area (%). (**G**) Scheme illustrating the drug actions of ProAgio in lung cancer.

## Data Availability

The raw data supporting the conclusions of this article will be made available by the authors on request.

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
