# Peer review of "Modulating Tumor Immunity by Targeting Tumor Fibrotic Stroma and Angiogenic Vessels for Lung Cancer Treatment"

_cancers, 2024, doi:10.3390/cancers16132483_

Round 1

Reviewer 1 Report

Comments and Suggestions for Authors

Manuscript "Modulating Tumour Immunity by Targeting Tumour Fibrotic Stroma and Angiogenic Vessels for Lung Cancer Treatment" is an intriguing manuscript that offers a knowledgeable approach for treating disease. The manuscript contains numerous ambiguities in certain sections that necessitate the authors to reframe and provide additional information.

1. The discussion section should include lines 67-85 in the introduction. Additionally, it would be beneficial for the authors to provide a schematic depiction of the entire mechanism.

2. Provide a brief explanation of PDL in the abstract and include the abbreviation.

3. What is the abbreviation for the DMEM medium? Kindly furnish.

4. The ELISA quantification assay should be referenced by the authors.

5. The authors should only provide an explanation of the outcomes of the specific experiment they conducted in the results section of "3.1. ProAgio Enhances Checkpoint Inhibitor Immunotherapy in Lung Cancer." The entire section should be rewritten, and the following statement should be redirected to the discussion section, as it is not a component of the specific study.

6. The authors have conducted a comparison between their study and the preceding one in the entire results section. This is resulting in a manuscript that is difficult to comprehend and is causing confusion. The entire result section should be rewritten. The significance of the studies should be compared in the specific section if the discussion section is distinct.

7. The authors should include the future research aspect and the benefits of the study for human welfare in the discussion as the significance of the research is lacking.

8. The entire manuscript contains numerous statements that are repeated from the abstract. Abstracts ought to be distinctive.

In general, the study is commendable; however, the authors should reconsider the manuscript's framework.

Comments on the Quality of English Language

minor edit required

Reviewer 2 Report

Comments and Suggestions for Authors

The manuscript “Modulating Tumor Immunity by Targeting Tumor Fibrotic Stroma and Angiogenic Vessels for Lung Cancer Treatment” by Yi Yuan et al shows interesting results in the field of cancer immunotherapy. There are some main concerns to be addressed before publication.

-       -The Aim of the study is not clearly described in the abstract and in the introduction section.

-    -In the introduction section the authors should better describe the concept concerning the CAF reduction associated with low PDL-1 expression (from line 77 to line 79).  

-Figure 1 describe the effects of ProAgio on inducing in vitro CAF apoptosis and on reducing CAF aggressive phenotype in A549 xenografts. One lung tumor model is not enough to demonstrate authors’ hypothesis, furthermore to better reproduce the in vivo tumor microenvironment, a mixtures of CAFs and cancer cell should be subcutaneously injected in nude mice to generate xenografts that may be a  suitable model to demonstrate the in vivo CAFs depletion by ProAgio.

Comments on the Quality of English Language

 Moderate editing of English language required

Reviewer 3 Report

Comments and Suggestions for Authors

Yuan et al. explore the impact of tumor fibrosis on lung cancer immunity, presenting an intriguing study with significant clinical implications. However, several areas require further clarification and enhancement:

1. The abstract should be rewritten to include quantitative data, providing a more precise summary of the study's findings and significance.

2. The number of mice or replicates in each experiment should be represented as individual data points (dots) rather than solid bar plots.

3. Statistical significance values should be provided to support the data analysis.

4. The specificity of ProAgio needs to be addressed in detail.

5. It is important to report whether mice develop any immune adverse reactions following repetitive administration of ProAgio.

6. The mechanisms by which ProAgio enhances αPDL1 delivery to lung cancer nodules should be explained.

7. The authors should indicate if they have tested other immune checkpoint inhibitors in their experiments.

8. The impact of combination therapy on T cells in lymphatics should be discussed.

9. A comprehensive conclusion should be provided, summarizing the key findings and suggesting future research directions.

Comments on the Quality of English Language

Good.

Round 2

Reviewer 1 Report

Comments and Suggestions for Authors

Now it seems fine. 

Comments on the Quality of English Language

Minor edit required 

Reviewer 2 Report

Comments and Suggestions for Authors

There are no additional comments for authors. They well addressed each points of the previuos review report. 

Comments on the Quality of English Language

Minor editing of English language required